# The Impact of Air Pollution on Asthma Severity among Residents Living near the Main Industrial Complex in Oman: A Cross-Sectional Study

**DOI:** 10.3390/ijerph21050553

**Published:** 2024-04-26

**Authors:** Souad Mahmoud Al Okla, Fatima Al Zahra Khamis Al Rasbi, Hawida Said Al Marhubi, Shima Salim Al Mataani, Yusra Mohammed Al Sawai, Hasa Ibrahim Mohammed, Muna Ali Salim Al Mamari, Salwa Abdullah Abdulrahim Al Balushi, Abdul Qader Abbady

**Affiliations:** 1College of Medicine and Health Sciences, National University of Science and Technology, P.O. Box 391, Sohar 321, Oman; fatima180592@nu.edu.om (F.A.Z.K.A.R.); hawida180498@nu.edu.om (H.S.A.M.); shima180339@nu.edu.om (S.S.A.M.); yasri180350@nu.edu.om (Y.M.A.S.); 2Department of Biology, Faculty of Sciences, Damascus University, Damascus P.O. Box 30621, Syria; 3Liwa Extended Health Center, Ministry of Health, Liwa 325, Oman; dr.hessaalbelushi@yahoo.com (H.I.M.); m.almamari@hotmail.com (M.A.S.A.M.); 4Falaj Al Qabail Health Center, Sohar 321, Oman; 5Division of Molecular Biomedicine, Department of Molecular Biology and Biotechnology, Atomic Energy Commission of Syria (AECS), Damascus P.O. Box 6091, Syria; aabady@aec.org.sy; 6Department of Biology and Medical Science, Faculty of Pharmacy, International University for Science and Technology (IUST), Damascus, Syria

**Keywords:** asthma, industrial pollution, air pollution, lung diseases, health survey, Oman

## Abstract

Background: Asthma is a widespread chronic respiratory disease that poses a significant public health challenge. The current study investigated the associations between air pollution and asthma severity among individuals residing near the Sohar industrial port (SIP) in Oman. Despite the presence of multiple major industrial complexes in Oman, limited knowledge regarding their impact on respiratory health is accredited. Hence, the primary objective of this study is to offer valuable insights into the respiratory health consequences of industrial air pollution in Al Batinah North. Methods: The state health clinics’ records for patient visits related to asthma were collected for the timeframe spanning 2014 to 2022. Exposure was defined as the distance from the SIP, Majan Industerial Area (MIA), and Sohar Industerial Zone (SIZ) to determine high-, intermediate-, and low-exposure zones (<6 km, 6–12 km and >12 km, respectively). Exposure effect modifications by age, gender, and smoking status were also examined. Results: The conducted cross-sectional study of 410 patients (46.1% males and 53.9% females) living in over 17 areas around SIP revealed that 73.2% of asthmatics were under 50 years old, with severity significantly associated with closeness to the port. Risk ratios were estimated to be (RR:2.42; CI95%: 1.01–5.78), (RR:1.91; CI95%: 1.01–3.6), and (RR:1.68; CI95%: 0.92–3.09) for SIP, MIP, and SIZ areas, respectively, compared to the control area. Falaj Al Qabail (6.4 km) and Majees (6 km) had the highest number of asthma patients (*N* 69 and *N* 72) and highest percentages of severe asthma cases among these patients (28% and 24%) with significant risk ratios (RR:2.97; CI95%: 1.19–7.45 and RR:2.55; CI95%: 1.00–6.48), correspondingly. Moreover, severe asthma prevalence peaked in the 25–50 age group (RR:2.05; CI95%: 1.26–3.33), and this linkage between asthma and age was much more pronounced in males than females. Smoking and exposure to certain contaminants (dust and smoke) also increased the risk of severe asthma symptoms, but their effects were less important in the high-risk zone, suggesting much more important risk factors. A neural network model accurately predicted asthma risk (94.8% accuracy), with proximity to SIP as the most influential predictor. Conclusions: This study highlights the high asthma burden near SIP, linked to port proximity, smoking, and wind direction as major risk factors. These findings inform vital public health policies to reduce air pollution and improve respiratory health in the region, prompting national policy review.

## 1. Introduction

Air pollution is a substantial environmental issue with profound implications for public wellbeing, particularly respiratory health. Industrial air pollution is a significant concern, with numerous studies highlighting its adverse effects on respiratory health [1,2,3]. Industrial activities, such as the combustion of fossil fuels, emit a wide range of pollutants, among which fine particulate matter (PM2.5) is prominent. These pollutants have been found to possess detrimental effects on respiratory health, contributing to the development of symptoms, exacerbating asthma, and increasing the risk of chronic respiratory diseases [4,5]. Moreover, long-term exposure to air pollution increases the risk of respiratory infections, cardiovascular diseases, and premature mortality [6,7,8]. It is noteworthy that living near industrial complexes augments exposure to air pollutants, elevating the risk of respiratory diseases such as asthma, chronic obstructive pulmonary disease (COPD), and lung cancer [9,10].

Asthma, a chronic respiratory condition affecting 339 million people globally, includes severe symptoms in 5–10%, leading to approximately 461,000 deaths annually, with a significant impact on low- and middle-income countries [11,12,13,14,15]. In Oman, asthma prevalence is 7.3% among adults and 12.7% among children, with high hospitalization rates and reliance on rescue medications [16,17,18]. Asthma management in Oman falls below Global Initiative for Asthma (GINA) guidelines, with substantial economic costs [16,18].

Asthma is a multifactorial disorder influenced by both genetic and environmental factors which drastically contribute to its development and progression [19]. Even though genetic susceptibility plays a crucial role, the development and progression of asthma also depend profoundly on numerous interactions with various environmental triggers [20]. Several issues, including family history of asthma, specific genetic variants related to the immune system function, and airway responsiveness, have been identified as risk factors [21]. In the same vein, epigenetic factors such as DNA methylation, histone modifications, and non-coding RNAs are also important contributors to the development and progression of asthma [22,23]. However, the increasing prevalence of asthma is primarily attributed to environmental changes, as genetic and epigenetic factors alone are unlikely to undergo significant alterations within a short timeframe [24,25]. Extensive epidemiological and clinical experimental investigations consistently demonstrate a compelling association between air pollution exposure and the increased risk of exacerbating asthma symptoms. These studies have consistently shown a clear link between poor air quality and increased asthma incidence in both children and adults [26,27,28]. Furthermore, long-term exposure to various types of ambient air pollution, such as traffic-related air pollution (TRAP) and indoor pollutants, has been identified as a significant factor in asthma development [29,30]. Furthermore, growing evidence suggests that air pollution not only triggers asthma exacerbations but also contributes to the development of new-onset asthma cases [10,31].

Several previous studies have emphasized the growing prevalence of asthma in low- and middle-income countries experiencing transitions, notably China, India, Brazil, and Oman. These studies highlight the importance of multiple factors, including rapid urbanization, changing lifestyle habits, and increased exposure to environmental allergens, as key contributors to the escalating burden of asthma in these regions [32,33,34,35].

The rapid industrialization and urbanization in Oman have led to a significant escalation in air pollution levels, primarily driven by industries such as oil refineries and manufacturing facilities [17]. The SIP, one of the largest in the Middle East, has witnessed rapid industrial development, resulting in the emission of several harmful substances [36,37]. Recent reports show heightened levels of particulate matter (PM10), nitrogen dioxide (NO_2_), nitrogen oxides (NOs), sulfur dioxide (SO_2_), and volatile organic compounds in the surrounding area [36,37]. Research by Al-Wahaibi and Zeka [38] also found increased pollutant concentrations near SIP, exceeding national standards. These contaminants contribute to poor air quality in the surrounding area, posing potential risks to both the environment and human health [36,39].

Despite substantial industrial growth at SIP, research on asthma severity in nearby residents is scarce. Existing studies mainly focus on asthma development, not severity. Moreover, the combined impact of air pollution from SIP, SIZ, and MIZ has not been explored. Furthermore, the influence of wind direction on the port’s surroundings has not been studied. Therefore, this study aims to assess the relationship between air pollution and the severity of asthma symptoms among residents near the port to bridge the knowledge gap and understand the potential consequences of cumulative air pollution on asthma severity within major industrial complexes. The findings of this study shall enable policymakers, regulators, and public health authorities to develop targeted interventions to mitigate the detrimental effects of air pollution on respiratory health.

## 2. Methods

### 2.1. Study Population

The current study was carried out in the rapidly growing SIP area, a densely populated region in the Sultanate of Oman. Residents living in close proximity to the major industrial complex were considered the study population; it is estimated at about 108,274 people, according to the latest census. A sample of 410 asthma patients (46.1% males and 53.9% females, with the majority, ~73.2%, under 50 years old) from 17 different areas surrounding SIP was selected based on the locations of healthcare centers serving asthma patients from nearby regions. These centers, chosen for their proximity to SIP, aimed to capture a representative sample of those affected by industrial pollution. Patient numbers varied across centers, reflecting population density differences. Additionally, data from regions beyond a 15 km radius were consolidated into a single control group for comparative analysis, ensuring geographic diversity and representation in our sample.

Medical data of the asthma patients were obtained from the national Al-Shifa electronic health recording system, authorized by the Omani Ministry of Health (MoH). Three healthcare centers located in Nabr, Falaj Al Qabail, and Liwa were selected for data collection due to their proximity to the industrial complex and significant asthma patient records. Eligible participants included adults diagnosed with asthma who provided signed informed consent and resided within Al Batinah North Governorate for at least 1 year without prolonged stays elsewhere. Exclusions comprised individuals residing outside the Governorate, pregnant women, and those unwilling to sign consent forms. Additionally, healthy individuals with chronic diseases, other respiratory conditions, or comorbidities potentially confounding study results, as well as regular travelers between polluted and unpolluted regions and those with important missing data, were excluded.

### 2.2. Asthma Severity Assessment

Qualified physicians conducted a thorough evaluation of the participants showing asthma symptoms to determine the severity classification, and the assessment was in adherence to the standardized criteria established by the Global Initiative for Asthma (GINA) [40]. This assessment was performed prospectively during patients’ health center visits, overseen by qualified physicians. Classification relied on a comprehensive evaluation, including medical record review and symptom assessment. By considering the type and frequency of reported asthma symptoms, the physicians categorized the severity into three distinct levels: severe, moderate, and mild. The classification framework provided insight into the range of asthma severity among participants. This enabled a meaningful analysis of how air pollution impacts their symptoms.

### 2.3. Exposure Zones

Asthma patients were categorized into three exposure zones based on their closeness proximity to SIP. The high-exposure zone, situated within a <6 km radius of the port, experiences high levels of air pollutants associated with asthma exacerbations [29,41], resulting in increased symptoms, hospitalizations, and emergency department visits [41]. Located between within a range of 6 and 12 km from the port, the middle-exposure zone still poses significant health impact risks, including aggravated asthma symptoms, decreased lung function, and a higher prevalence of asthma [42]. Areas beyond 12 km from the port constitute the low-exposure zone, characterized by comparatively lower levels of air pollution, but may still experience health effects during periods of elevated pollution from industrial activities or weather-related events [43].

### 2.4. Sample Size Determination

The sample size in this study (*N* = 410) was larger than estimated (*N* = 246) using standard sample size equation, based on the estimated asthmatic population in Sohar (~16,000 patients) and the proportion of severe asthma among asthma patients (20%), desired confidence (95%), and the margin of error (5%) levels. Our estimation, based on prior research, suggested a 20% prevalence of severe asthma among Omani asthmatic patients, ensuring a representative sample size. Despite encountering missing data for variables like family size and marital status, our study included a robust analysis. Notably, significant missing data for these variables led to their exclusion from the study.

### 2.5. Data Collection

Data for this study were collected from three state health centers located in the study provinces, utilizing the national Al-Shifa electronic health recording system. The system provided extensive data from 2014 to 2022, ensuring a broad temporal scope. Data security and patient privacy were of utmost importance, and the research team strictly adhered to relevant regulations, approvals, and data sharing agreements throughout the data collection process to ensure integrity and confidentiality. Medical records were utilized to gather clinical data related to asthma severity, medical history, and demographic information (gender, marital status, place of residence, number of years of residency, smoking status).

### 2.6. Statistical Analysis

Comparisons between groups were performed using a Chi-squared test for categorical outcomes and a Mann–Whitney U-test between categorical and nonparametric numerical data. Risk ratios (RRs) and the 95% confidence intervals (CIs) were calculated for the different exposure zones and factors in this study. Descriptive statistics, data analysis, and artificial neural network (ANN) model fitting were carried out with SPSS v22 (IBM Corp., Armonk, NY, USA), the graphs were created using GraphPad Prism^®^ 7 (GraphPad Software, San Diego, CA, USA), and the level of two-sided significance of a *p*-value was <0.05. Average asthma values (severe = 3, moderate = 2, and mild = 1) were calculated and represented in the different bar graphs, error bars indicated the standard error of the mean (SEM), and the number of patients was always shown above each bar. The significance of the differences was determined by a Mann–Whitney U-test at *p* < 0.05 * and *p* < 0.005 **.

### 2.7. Ethical Considerations

Ethical approval and data sharing agreements were obtained from the appropriate authorities, including both the Ministry of Health and the Ethics and Biosafety Committee of the College of Medicine & Health Sciences at the National University of Science and Technology. All data were de-identified and handled in compliance with data protection regulations to ensure patient confidentiality and privacy.

Given the retrospective nature of this study and the utilization of de-identified data from the Shefa system, where patient identities were rigorously safeguarded, the acquisition of individual consent was considered unnecessary.

## 3. Results

### 3.1. Distribution of Asthma Severity among Omani Patients

The distribution of asthma severity among 410 patients (46.1% males and 53.9% females) living in 17 different areas surrounding SIP was investigated in the current study. The majority of asthmatics were under 50 years old (73.2%), with severity significantly associated with the proximity to the port. Most cases were classified as moderate (53.2%), while severe (20.5%) and mild (26.3%) asthma rates were relatively similar among both male and female patients (Figure 1). Differences between moderate and other asthma severity levels were highly significant according to the one-sample Chi-square statistical test.

Asthma patients were sorted into three zones representing different pollution exposure levels based on their geographic location and distance from SIP (Table 1). Data analysis revealed a notable clustering of asthma cases in the propinquity to SIP (51.2%), indicating a higher prevalence of the condition within these specific areas. Additionally, an increase in severe asthma rates as the distance to SIP decreased was observed from low (13%), to middle (35.2%), to high (51.9%), suggesting a relationship between proximity to SIP and asthma severity. This pattern was consistent among both male and female patients, with no statistically significant differences found between the two genders.

A comprehensive analysis was performed to calculate the average asthma severity among patients residing in the three exposure zones. The analysis considered the entire patient population as well as distinct groups based on gender (males and females) and the average of distances covered by these patients within each of the three exposure zones (Figure 2). The results showed a clear decrease in the average asthma severity in the low-exposure zone compared to the other two zones, which had similar average severities and margins of error. Although the average of asthma severity for male and female patients in the middle-exposure zone was similar, males had a slightly lower average severity in the high-exposure zone, while females had a lower average severity in the low-exposure zone.

We conducted an in-depth analysis to investigate the relationship between asthma severity, gender, and proximity to SIP. This involved comparing the average distances from patients’ residences to the port for severe asthma cases with those for a combined group of moderate and mild cases (Figure 3). We confirmed a strong association between severe asthma and proximity to SIP compared to combined asthma (6.25/7.88 km, *p* = 0.071); this difference was more pronounced and significant for males (*p* = 0.035) than for females (*p* = 0.502). On the other hand, male patients with severe asthma were significantly closer to SIP than females (5.33/6.91 km, *p* = 0.033), but no such difference was observed between males and females with combined moderate and mild asthma (~7.8 km, *p* = 0.181).

### 3.2. Industrial Zones in Sohar and Severe Asthma

In addition to SIP (latitude: 24.4904, longitude: 56.615), other industrial areas in Sohar’s northern region, such as SIZ (24.3813, 56.6516) and MIA (24.4322, 56.5728), were also investigated in terms of their contribution to severe asthma risk. Patients were divided into high-, middle-, and low-exposure zones around each industrial area, and a control area (CA, 24.5103, 56.5157) located to the north-west of Sohar was selected for comparison (Table 2). A comparative analysis was conducted to assess the prevalence of severe asthma among patients with different forms of asthma, specifically in high- and low-exposure areas surrounding industrial zones. Our results demonstrated that severe asthma was most strongly associated with SIP, followed by MIA, when compared to SIZ or the control area. Furthermore, our findings showed statistically significant differences for both SIP and MIA. Significant statistical differences were observed between the compared groups for SIP and MIA, establishing the substantial impact of these industrial zones.

This study also investigated the relationship between exposure to SIP pollution and the development of severe asthma for patients in the northern, southern, and middle regions surrounding the port (Table 2). Patients in the intermediate category are precisely situated within a latitude range that coincides with SIP’s location, encompassing a width of 5 km. Notably, patients located at longitude lines below 56.5 were intentionally excluded from the calculation. A comparison was made between the frequency of severe asthma cases and other forms of asthma for patients in the middle, northern, and southern regions. Interestingly, the central region exhibited the highest level of risk in contrast to the northern region, implying a greater degree of vulnerability. In a fascinating observation, the southern region displayed a comparable level of hazard to that of the port itself, as evidenced by the risk ratio between these two areas.

### 3.3. Industrial Areas and Prevalence of Severe Asthma

The geographical distribution of Omani asthma patients around SIP and other industrial areas was mapped based on latitude and longitude coordinates of each geographical region (Figure 4). By comparing these coordinates with those of the industrial areas, a significant increase in severe asthma patients was found in the proximity to SIP and other industrial regions, particularly in Ghadfan (27%, 2.6 km from SIP), Falaj al-Qabil (28%, 6.4 km from SIP), and Majees (24%, 6 km from SIP). Furthermore, the risk ratio in these regions was considerably high, measuring 2.88, 2.97, and 2.55, respectively, compared to other areas. Additionally, the differences in severe asthma cases compared to the control condition (moderate and mild asthma in areas more than 12 km away from SIP) were statistically significant, as determined by the Chi-square test (Table 3).

### 3.4. Age-Related Patterns in Asthma Severity

The prevalence of asthma across distinct age groups was examined and compared between total and high-exposure areas (<6 km from SIP) for both males and females (Table 4). Our findings consistently reveal a prevalence of asthma in the young and middle-aged groups, each accounting for approximately 36.6% of cases. This pattern remains consistent in the high-exposure zone, with a slight increase to approximately 39%. Notably, severe asthma represents the highest percentage within the middle-aged group, comprising around 27% of cases and approximately 50% of severe asthma instances.

Furthermore, the middle-aged group exhibits a two-fold higher risk of severe asthma compared to the youth group (<25), and this difference proves statistically significant. The risk ratio slightly decreases to 1.78 in the high-exposure zone, yet the statistical significance remains intact. When comparing gender, males exhibit a higher risk ratio (2.8) for severe asthma within the middle-aged group. This difference persists regardless of the total condition or the area most exposed to pollution, with statistical significance observed in the total condition. Conversely, the elder group (>50) shows a relatively lower risk compared to the aforementioned age group, and the risk further decreases in the area with the highest pollution exposure (Table 4).

The severity of asthma across different age groups and genders was examined to gain insights into its patterns. A comparison of the average asthma severity revealed distinct variations among these groups (Figure 5). Notably, asthma severity tended to increase in the middle-aged group (25–50 years) when compared to both young (<25 years) and elder (>50 years) groups, and these differences were statistically significant. Interestingly, within the area with high pollution exposure (<6 km from SIP), the levels of significance between these age groups seemed to decrease. However, a substantial difference was noticed in the average asthma severity of males in the youth group, as compared to the middle-aged and elder ones. This difference was less prominent among females. Proximity to SIP did not appear to have a clear impact on the distribution of asthma severity across the different age groups, but seemed to reduce the differences between these groups.

An analysis of the average ages of patients according to their asthma severity levels revealed a direct relationship between the severity of asthma and the average patient ages, and the difference between the patients with severe and mild asthma was statistically significant (*p* = 0.02) (Figure 6). Interestingly, this relationship was distinctly apparent and statistically significant among males (severe/moderate *p* = 0.033, severe/mild *p* < 0.001, and moderate/mild *p* = 0.002), whereas it was nearly absent among females. Moreover, significant differences in the average ages across two asthma severity categories (moderate *p* = 0.001 and mild *p* < 0.001) were observed between males and females. Surprisingly, proximity to the port seemed to diminish the association between asthma severity and age, and most of the significant differences seemed to have been lost, except between severe and mild asthma in males, which was reduced to *p* = 0.018. In the case of females, this relationship even appeared to be reversed.

### 3.5. Asthma Severity and Smoking Status

In this study, we conducted an analysis on the association between smoking and asthma severity, with a particular focus on the impact of the primary pollution source, namely SIP. Notably, our findings revealed a remarkably low percentage (5.4%) of smokers among the patients included in this study. Moreover, we observed a higher prevalence of smokers (10.1%) among male patients compared to (1.4%) among female patients (Table 5). When examining the distribution of asthma severity levels among patients based on their smoking status, smokers exhibited a higher percentage (90.9%, 20 cases) of severe and moderate asthma compared to passive smokers (80.5%, 33 cases) and non-smokers (72%, 249 cases). However, these percentages slightly decreased from smokers (88.1%, 8 cases), to passive smokers (81.8%, 18 cases), to non-smokers (71.5%, 128 cases) in the closest area to SIP with the highest pollution exposure (<6 km). Furthermore, when assessing the risk ratio associated with smoking, we found that smokers had higher chances at ~1.7 and 1.1 of developing severe asthma compared to mild asthma or combined asthma (mild and moderate), respectively, in comparison to non-smokers. These values decreased to 1.1 and 0.46 for patients residing in the highly exposed area.

A compelling pattern emerges when examining the average asthma severity among different groups of patients, namely smokers, passive smokers, and non-smokers. A discernible and consistent gradient can be observed in the association between the average asthma severity and smoking (Figure 7). However, it is noteworthy that this link becomes somewhat less distinct within the community of patients residing in close proximity to SIP. This finding implies that pollution stemming from the port may yield a more significant impact on the average asthma severity than smoking itself.

### 3.6. Asthma Severity and Air Contaminants

During the process of collecting patients’ data, their opinions were solicited regarding the primary pollutants they believed to occur in their surroundings, potentially triggering their asthma symptoms (Table 6). The identified pollutants included dust, incense, perfume, and smoke. Among the surveyed asthmatic patients, 68.5% acknowledged the existence of pollution matter in their environment, which they considered a potential cause of their asthma symptoms. Interestingly, the percentage of severe asthmatics within this group (19.2%) was lower compared to those who were uncertain about the presence of a clear pollution factor in their surroundings (23.2%). Analyzing the four investigated pollutants, it was found that incense was the most frequently reported (33.4% of responses and 81.8% of respondents). However, incense had the lowest risk ratio (RR = 0.96) among the pollutants, indicating a lesser association with severe asthma (20.1% of patients). When considering different types of asthma and comparing them with the exposed and unexposed populations, smoke emerged as the most significant risk factor for the development of severe asthma, with a risk ratio of 1.27. Dust followed closely with a risk ratio of 1.2, while perfume exhibited a risk ratio of 1.11. It is noteworthy that the risk ratios for smoke (1.41) and perfume (1.21) slightly increased in the area closest to the port (within a 6 km radius), whereas the one for dust decreased (1.02). Although the percentage of respondents who identified smoke as a contributing factor was the lowest (29.6%), it also collected the lowest affirmative responses (“yes”) regarding its presence as a pollutant (12.1%).

Subsequently, we conducted a thorough investigation of the disparities in “Yes” responses in the contaminant survey between males and females, as well as the existence of a relationship with severe asthma and proximity to the port (Figure 8). The findings clearly indicated a statistically significant divergence of opinions regarding the negative impact of these pollutants on asthma development (*p* < 0.001), except for dust and perfume, where no significant differences were observed. Specifically, when comparing the percentage of male and female “Yes” voters, smoke emerged as the least significant pollution factor among those with high statistical significance. It is noteworthy that there was an unanimous agreement, statistically significant, that incense was the most bothersome factor for male and female asthma patients. This perception remained consistent regardless of the severity of asthma or proximity to SIP. Additionally, an interesting observation was the increase, particularly among males, in affirming the presence of smoke contamination among the group of severe asthma patients and those living near SIP. Although not reaching full statistical significance, there was a notable difference of opinion on the effect of smoke between males and females in these three conditions, approaching significance (*p* = 0.06 by the Chi-square test).

### 3.7. Predicting Asthma Risk Using Artificial Neural Networks

A multilayer perceptron neural network was employed to construct a predictive model for a dependent variable, which represents the risk of developing asthma. The model was developed based on a set of predictor variables, namely age, smoking status, region, gender, and proximity to the three industrial zones. In this study, the dependent variable, referred to as “Asthma Risk (AR)”, was categorized into five levels ranging from 1 (Very Low Risk) to 5 (Very High Risk). To ensure homogenous comparisons, the covariates were rescaled using the standardized method as the default approach. To apply the neural network technique effectively, the dataset was partitioned into two distinct sets: a training set comprising 70% of the data (*N*= 286) and a testing set comprising the remaining 30% (*N* = 124). It is important to note that there were no missing values in either set. The training set was utilized to train the neural network and develop the model, while the testing set served as an independent dataset to monitor errors during training and prevent overfitting (Figure 9). For the construction of the AR network, a single hidden layer was employed, with the number of units in the layer architecture ranging from 1 to 50 [44].

The network stopping rule was implemented to terminate training when the maximum number of steps was reached without observing a decrease in error. The model achieved a 5.2% rate of incorrect predictions with a training time of 17 milliseconds and a cross-entropy error of 52. Furthermore, the model’s performance was assessed again, yielding a percentage of incorrect predictions of 9.7% with a cross-entropy error of 45. The classification table displays the percentages of correct predictions, revealing an overall accuracy of 94.8% for the training dataset and 90.3% for the testing dataset. It is worth noting that the correct prediction percentages for risk levels were higher for average risk compared to very high or very low levels (Table 7).

We examined the sensitivity and specificity (represented as 1—the false positive rate) of the model in predicting each level of the AR model, using the combined training and testing samples. To assess the performance of the model, we employed an area under the curve (ROC) analysis on the combined training and testing samples. Remarkably, the ROC curve for all predicted AR levels surpassed a threshold of 0.98, indicating highly acceptable results.

Additionally, an assessment of the normalized importance of each predictor variable revealed the prominence of proximity to SIP as the most influential factor. Subsequently, several covariates such as proximity to the MIA and SIZ areas, age, and regions followed in terms of importance (Figure 10). Conversely, smoking and gender had minimal impact on the constructed neural network model. Normalized importance serves as a quantifiable measure of the extent to which the predicted value of the dependent variable, AR, would be affected by the exclusion of a particular predictive indicator.

## 4. Discussion

The fast industrialization and expansion of Oman’s SIP have led to a continuous growth in industries, resulting in increased air pollution. Addressing this pressing issue necessitates the undertaking of a comprehensive study. This contemporary cross-sectional study aims to investigate the association between industrial air pollution from SIP in Oman and respiratory health outcomes among individuals visiting health centers located in close vicinity to the port. Thus, our objective was to assess the prevalence and severity of asthma among a sample of 410 Omani asthma patients (46.1% males, 53.9% females). Most patients were under 50 years old (73.2%). Asthma severity distribution was as follows: moderate (53.2%), severe (20.5%), and mild (26.3%). Proximity to the port significantly influenced severity, with 51.2% residing within 6 km (high-exposure zone). Severe asthma rates increased closer to the port, while the lowest severity was observed in the low-exposure zone (>12 km). The middle- (6–12 km) and high-exposure zones had similar severity levels.

The results of this study show a close relationship between exposure to air pollution from SIP and other industrial areas and the severity of asthma among patients living nearby. The percentage of severe asthma cases was higher among patients living closer to the port (<6 km), and decreased as the distance from the port increased. This indicates that proximity to the port is a critical risk factor for developing severe asthma. These findings align with previous studies. Al-Wahaibi and Zeka [38] also found that living closer to SIZ increased the risk of asthma and other respiratory conditions. Other studies have linked industrial air pollution to worse asthma symptoms due to irritants in the emissions [45,46]. Moreover, Mock et al. highlighted the significant impact of city industrial zoning on pediatric asthma outcomes in communities with limited access to air monitoring [47].

The current study’s findings are also in agreement with various international studies that found a strong association between living near ports or industrial sites and bad asthma outcomes [48,49]. For instance, a study by Guarnieri and Balmes [10] revealed that exposure to air pollutants, such as particulate matter and nitrogen dioxide, was associated with exacerbated asthma symptoms. Similarly, a study by Orellano et al. [50] found that exposure to traffic-related air pollution increased the risk of asthma exacerbation in children. This supports the notion that industrial pollution can play a vital role in exacerbating asthma and highlights the importance of considering the impact of proximity to industrial ports when examining asthma prevalence and severity.

Our study’s findings provide further support to the body of evidence that establishes a clear link between industrial exposure and heightened occurrences of severe asthma. Specifically, we observed that areas in close proximity to the MIA and SIP, characterized by significant exposure, exhibited notably increased rates of severe asthma. The risk ratios associated with these high-exposure zones were found to be 2.88 and 2.97, respectively. These results align with previous research conducted across diverse global regions, reinforcing the well-documented association between environmental pollution and respiratory diseases such as asthma.

Several research studies have shed light on the detrimental effects of air pollution on human health. Notably, a study conducted in China in 2016 revealed a strong correlation between air pollution resulting from industrial activities and the prevalence of asthma in children [51]. Similarly, a study conducted in the United States demonstrated that individuals residing in close proximity to industrial zones had elevated risks of developing respiratory diseases, including asthma [52].

Despite the lack of specific data from Oman, these international studies provide a comparative perspective. The current findings fill a significant gap in the literature by providing localized data from Oman. These results highlight the importance of addressing industrial pollution in Oman, particularly near high-exposure zones such as the MIA and SIP, to mitigate the health risks associated with such exposure.

However, the southern region exhibits a comparatively lower risk ratio of 2.55, albeit still elevated, warranting a comprehensive inquiry. This divergence may be attributed to variations in industrial activities, wind patterns, or other environmental and demographic factors that could potentially modulate the dissemination and impacts of pollutants, necessitating thorough examination. These findings add to the growing body of evidence substantiating the link between industrial pollution and severe asthma. They underscore the need for effective environmental regulations and public health measures, especially in high-risk areas.

The finding of higher risks in the central region closest to SIP also aligns with the results of Raaschou-Nielsen et al. [53], who found the strongest associations between traffic-related air pollution exposure and childhood asthma hospital admissions within the first 500 m from major roads. The fact that the southern region further from the sources still experienced an increased risk ratio of 2.55 compared to the reference population suggests that the impact of industrial emissions may extend over a wider area, though risks do appear to reduce with distance from the source.

Notably, among male patients, the high-exposure zone showed a slightly lower average of severity compared to the middle-exposure zone. Conversely, female patients demonstrated a lower average of severity in the low-exposure zone. Severe asthma patients lived closer to the port (average 6.25 km) than moderate/mild patients (average 7.88 km). This was more significant for males (severe: 5.33 km, moderate/mild: 7.8 km). To provide further details, we compared our findings with results from studies conducted in the vicinity of SIP, as well as in other countries [7,38,47,54,55,56,57,58].

Our results showed a clear decrease in the average asthma severity in the low-exposure zone compared to the other two zones, which had similar averages of severity and margins of error. These findings are consistent with previous studies conducted near SIP, where low-exposure areas exhibited lower asthma severity compared to high-exposure areas [38].

Furthermore, our examination of gender-specific differences revealed that, within the middle-exposure zone, the average asthma severity was similar for male and female patients. However, in the high-exposure zone (within 6 km from the port), females exhibited slightly higher asthma severity compared to males. It is important to note that our study did not establish a general trend of men having more severe asthma than women. These gender-related findings align with previous international studies that have reported similar patterns of asthma severity based on gender and exposure levels which could be due to variations in occupational exposures, horizontal segregation in the workforce, and diverse behaviors within the workplace, which can impact the level of exposure and subsequent health outcomes [59,60,61].

In addition, although the risk ratio for severe asthma was high for males across all three exposure zones, there was no universally established consensus that men had more severe asthma than women living away from industrial areas. Various lifestyle factors influence the severity of asthma. Men working in occupations with higher exposure to irritants or allergens may be more prone to asthma symptoms. Population-based studies in South Finland have shown a link between occupational exposures and prevalent asthma. Clinical evidence suggests that females experience increased asthma symptoms during puberty due to hormonal changes. Estrogen increases airway inflammation, while testosterone decreases it, as demonstrated in animal studies [62]. Occupational and lifestyle exposures contribute to the observed differences in asthma severity between men and women [63,64]. Additionally, individual sensitivity and the amount of irritants present can vary, leading to varying degrees of asthma symptoms [65].

Our analysis aimed to examine the relationship between asthma severity, gender, and proximity to SIP. While there was a trend suggesting that individuals with severe asthma resided closer to SIP compared to those with moderate and mild asthma, this association did not reach statistical significance for the overall population (6.25 km vs. 7.88 km, *p* = 0.071). However, a gender-specific analysis revealed that males with severe asthma exhibited a significant association with residing closer to SIP (5.33 km vs. 6.91 km, *p* = 0.033), while no such difference was observed for females (*p* = 0.502). Our study’s findings align with previous research indicating that men may be more susceptible to the adverse health effects of air pollution [66,67]. However, it is important to acknowledge the nuanced nature of this relationship. For instance, while some studies, such as the Dutch cohort study, found no significant differences in effects between men and women [68], others, like those by Miller et al. [69] and Kan et al. [66], have suggested that women might be more susceptible to these effects [70,71]. These disparate findings underscore the multifaceted nature of the interaction between air pollution exposure and respiratory outcomes. Biological disparities between genders and variations in the lung deposition of particles could play a role in shaping these differential responses [72,73,74]. Additionally, it is crucial to consider potential confounding factors such as socioeconomic status or smoking habits, which may influence both residence location and asthma severity, thus potentially confounding study results. Notably, our findings that males with severe asthma reside closer to SIP raises inquiries into potential gender-specific environmental burdens impacting asthma severity. Further investigation with a larger sample size is crucial to fully elucidate the complex interplay between air pollution exposure, gender, and respiratory health outcomes.

In our study, severe asthma prevalence, compared to other levels of the disease, was the highest in patients aged 25–50 years (27.3%, 2.05 risk ratio). Furthermore, this middle age group had the highest percentage of severe asthma patients (48.8%) and average asthma severity (2.1) compared to other age groups. Studies have shown that around half of middle-aged asthma patients developed the condition in adulthood rather than during childhood [75,76]. The annual incidence of asthma among adults is estimated to be 0.5%, similar to childhood incidence, but it remains uncertain whether adult-onset asthma is the same as childhood-onset asthma [77]. Interestingly, previous studies demonstrated that asthma in adults progresses at a faster rate than in childhood. This could be due to the complexity of adult-onset asthma, which is characterized by a diverse range of symptoms and outcomes. Unlike childhood asthma, which often enters a period of remission, asthma in adulthood tends to be more severe and progressive [78,79].

Furthermore, our findings revealed that the elder group (>50 years) exhibited a lower level of severe asthma compared to the mentioned age range. Moreover, in areas with high pollution exposure, the risk decreased even further. A study on the natural history of asthma supported our findings, indicating that atopy is not a risk factor in the elderly age group [80]. Interestingly, older patients who develop asthma have a similar incidence rate to younger individuals (100 per 100,000) [81]. However, the severity of asthma tends to be more pronounced in the older age group due to poor lung function and fixed airway obstruction [82]. In contrast, a study by Al-wahaibi suggested that living near the exposure source increased the risk of asthma in individuals over 50, indicating greater vulnerability within this age group [38].

Our study revealed a high prevalence of incense use among both male and female asthma patients (33.4%), regardless of asthma severity or proximity to the source of indoor pollution (SIP). However, it is noteworthy that incense showed the lowest risk ratio (0.96 risk ratio) among the pollutants examined, indicating a weaker association with severe asthma [83]. These findings are consistent with previous research conducted on Omani children from two regions with varying asthma prevalence. The studies demonstrated that exposure to incense triggers asthma symptoms but does not have a significant association with the prevalence of current asthma [54]. In contrast, smoke emerged as the most significant risk factor for the development of severe asthma. In the present study, 5.4% of patients were smokers, mostly males (10.1% of all patients). Moreover, smokers had a higher risk of developing severe asthma (1.7 risk ratio) compared to non-smokers with mild asthma. The average asthma severity was the highest in smokers, followed by passive smokers and then non-smokers. This trend was less prominent near the port. This finding aligns with previous studies that have observed the detrimental effects of smoke originating from various sources, including environmental tobacco smoke, air pollution (such as wood smoke and particulate matter), industrial pollution, and vehicle emissions, on asthma symptoms and asthma risk [58]. The results of our study emphasize the importance of reducing exposure to smoke and implementing measures to mitigate its adverse effects on individuals with asthma [83].

In addition, considerable evidence suggests that social and environmental factors significantly influence asthma prevalence, including aspects like neighborhood environment, social cohesion, economic stability, and access to quality education and healthcare [66]. While previous research has explored the independent effects of air pollution and social determinants of health on asthma, few studies have examined their combined impact. Understanding this combined influence is crucial, especially as communities facing poverty often experience heightened pollution exposure, leading to increased asthma morbidity [84]. Moreover, minorities and individuals with lower socioeconomic status tend to suffer worsened asthma outcomes due to pollution disparities [85,86]. Despite their acknowledged importance, incomplete patient records hindered our ability to comprehensively analyze confounding factors like socioeconomic status, as missing data exceeded 30%.

A multilayer perceptron neural network was used to construct a predictive model for the risk of developing asthma. The model achieved an overall accuracy of 94.8% for the training dataset and 90.3% for the testing dataset. The areas under the ROC curve for all predicted risk levels exceeded a threshold of 0.98, indicating highly acceptable results. The analysis of predictor variables revealed that proximity to the SIP industrial zone exerted the greatest influence on asthma risk, followed by proximity to other industrial zones, age, and regions. However, smoking and gender had minimal impact on the neural network model. These findings are consistent with previous studies that demonstrated the effectiveness of various neural network models in accurately predicting asthma-related outcomes [87,88]. These studies consistently highlight the pivotal role of air pollution and meteorological variables as influential factors in asthma prediction. However, it is worth noting that the present study found that smoking and gender had minimal impact on the constructed neural network model, which is different from some previous research. For example, a study by Ho et al. [89] found that smoking status and gender were significant predictors of asthma risk. This discrepancy may be due to differences in the study population, especially the low percentage of smokers in Omani society, or the specific neural network architecture employed. In addition, our statistical analyses did not show a direct relationship between asthma severity and gender, which makes our network’s prediction model more reasonable. Future research could explore the use of alternative neural network architectures or additional predictors to further improve the accuracy of these models. Our study makes significant scientific contributions by providing novel insights into the relationship between industrial pollution exposure and asthma severity in Oman. By identifying localized pollution disparities and demographic susceptibilities, we contribute to a deeper understanding of the complex interactions between environmental factors and respiratory health. However, limitations such as the use of cross-sectional data and the focus on a specific demographic and geographic area may affect the broader applicability of our findings. Despite these shortcomings, our research provides valuable insights that can inform future studies and interventions aimed at mitigating the impact of pollution on respiratory health.

## 5. Conclusions

The findings of this study provide several new insights into the relationship between industrial pollution exposure and asthma severity in Oman. Our study found higher rates of severe asthma north and south of SIP compared to the west, indicating localized differences in pollution exposure. This underscores the need for localized assessment of pollution emissions to understand the varying burden of asthma severity around SIP and other industrial zones. Moreover, middle-aged men face a higher risk of severe asthma compared to other age groups, particularly when compared to women in the same age range. This highlights the need for targeted interventions to address pollution-related health effects in specific demographic groups. This finding is novel and not previously reported in Oman or the literature. Furthermore, smoking worsens asthma, but air pollution’s severe impact near pollution sources may overshadow the effects of smoking on asthma severity. The interaction between smoking and air pollution is complex, with their health burdens not simply adding up. Finally, our study emphasizes localized pollution controls for asthma near pollution sources. Regional controls may be ineffective due to high-exposure “hotspots”. Recommendations include localized monitoring and mitigation in Oman. Therefore, our findings point to localized differences in impacts around pollution sources, an increased vulnerability in certain groups like middle-aged men, interactions between air pollution and smoking, and the need for localized pollution controls to maximize health benefits. By highlighting these localized aspects, this research helps to build a more nuanced understanding of asthma and pollution in this region.

## Figures and Tables

**Figure 1 ijerph-21-00553-f001:**
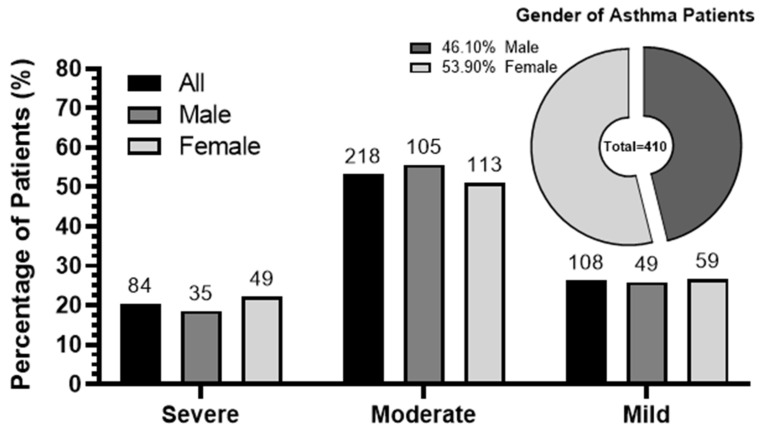
Asthma severity percentages in the population and gender distribution. Percentages (%) of patients by asthma severity level (severe, moderate, and mild) in the whole population (all) or clustered by gender.

**Figure 2 ijerph-21-00553-f002:**
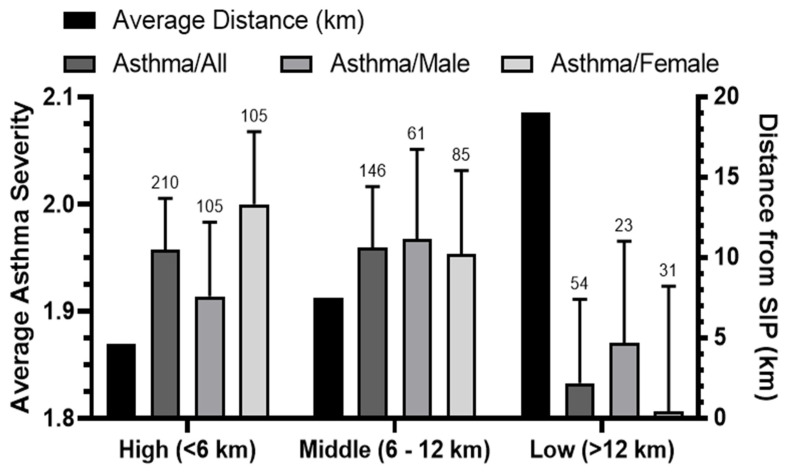
The relationship between exposure zones and the average of asthma severity. The average asthma severity of the studied population is depicted in relation to the exposure zones that were categorized as follows: high (<6 km), middle (6–12 km), and low (>12 km). To calculate the averages, asthma severity levels were represented by numerical values: severe = 3, moderate = 2, and mild = 1. The average distance (km) for the locations of these patients within each of the three exposure zones is shown.

**Figure 3 ijerph-21-00553-f003:**
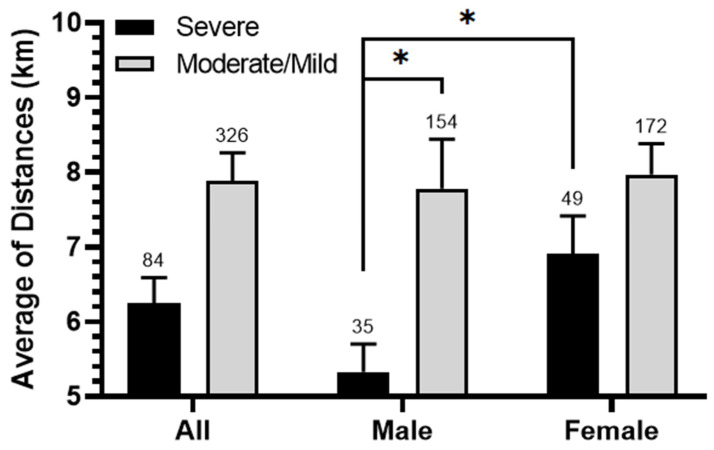
The relationship between the severity of asthma and the average distance of patients’ residences from SIP. Average distance (km) was calculated for male, female, and entire populations (all) and categorized by different levels of asthma severity (severe, moderate/mild). The significance of the differences was determined by a Mann–Whitney U-test at *p* < 0.05 *.

**Figure 4 ijerph-21-00553-f004:**
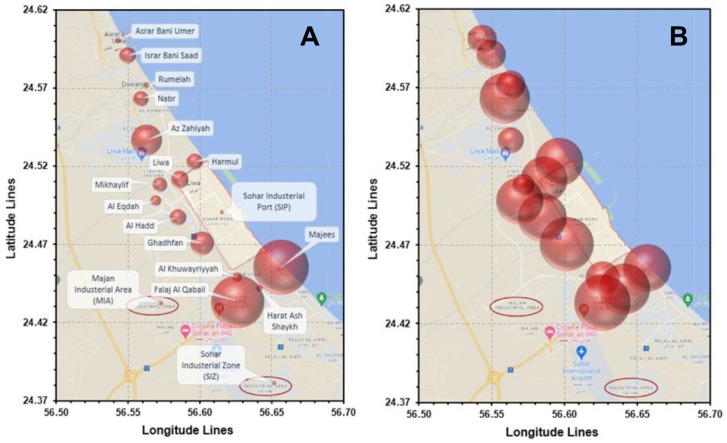
Comprehensive map representing the dispersion of asthma patients around SIP. The map illustrates various regions where the patients reside. Each region is portrayed by a circle, the size of which corresponds to either the total number of patients (**A**) or the percentage of severe asthma cases (**B**) within that particular region. Additionally, the map highlights the placement of MIA and SIP.

**Figure 5 ijerph-21-00553-f005:**
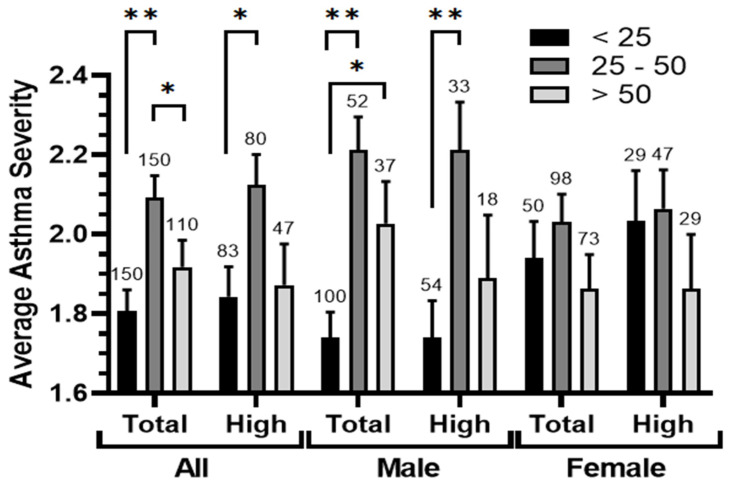
The relationship between the age ranges and the average asthma severity. Average asthma severity was calculated in the studied population (all, male, and female) in two location conditions (total and high-exposure zone <6 km from SIP) after clustering according to ages (<25, 25–50 and >50 years). The significance of the differences was determined by a Mann–Whitney U-test at *p* < 0.05 * and *p* < 0.005 **.

**Figure 6 ijerph-21-00553-f006:**
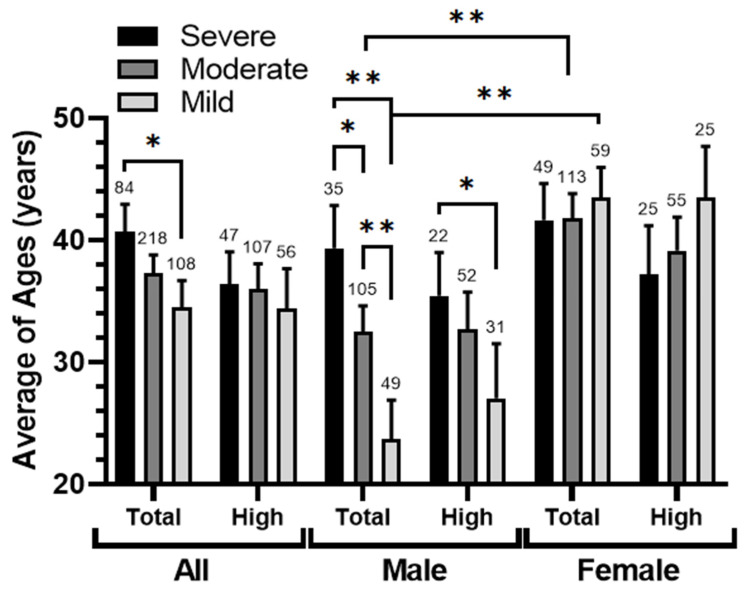
The relationship between asthma types and the average patient ages. Average patient ages in the studied population (all, male, and female) in two locations (total and high-exposure zone < 6 km from SIP) clustered according to asthma level (severe, moderate, and mild). The significance of the differences was determined by a Mann–Whitney U-test at *p* < 0.05 * and *p* < 0.005 **.

**Figure 7 ijerph-21-00553-f007:**
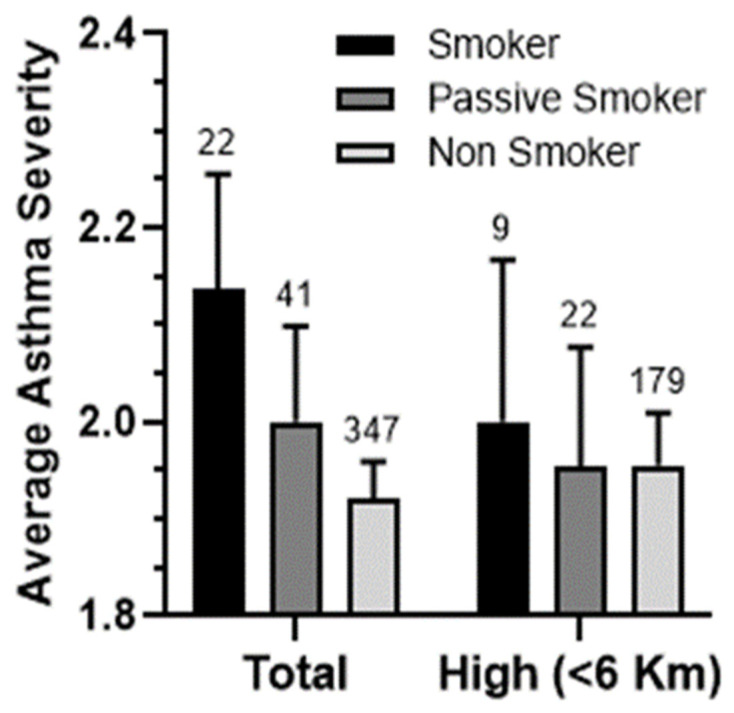
The relationship between patients’ gender, smoking status, and average asthma severity. The average asthma severity was calculated based on their smoking status: smokers, passive smokers, and non-smokers in the household.

**Figure 8 ijerph-21-00553-f008:**
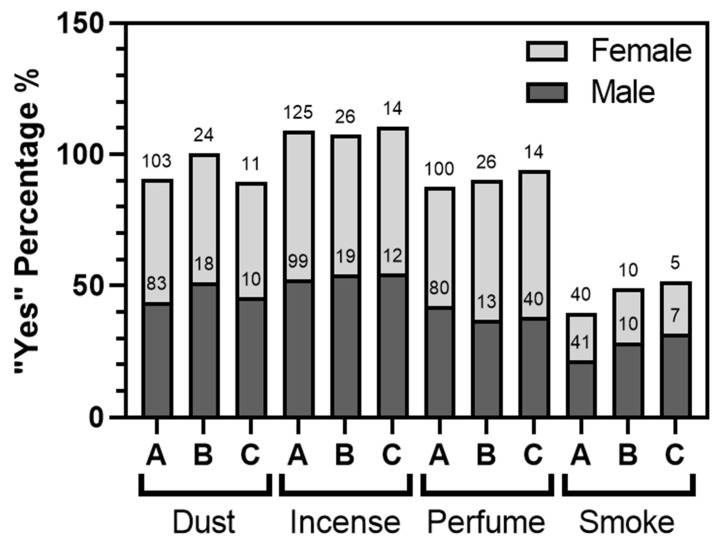
Relationship between contaminant exposure, asthma, and port proximity. Comparison of the percentages of “Yes” responses from the survey conducted on exposure to contaminants, particularly dust, incense, perfume, and smoke. The analysis is conducted for three groups: all patients (**A**), those with severe asthma (**B**), and those with severe asthma living within a 6 km radius of the port (**C**).

**Figure 9 ijerph-21-00553-f009:**
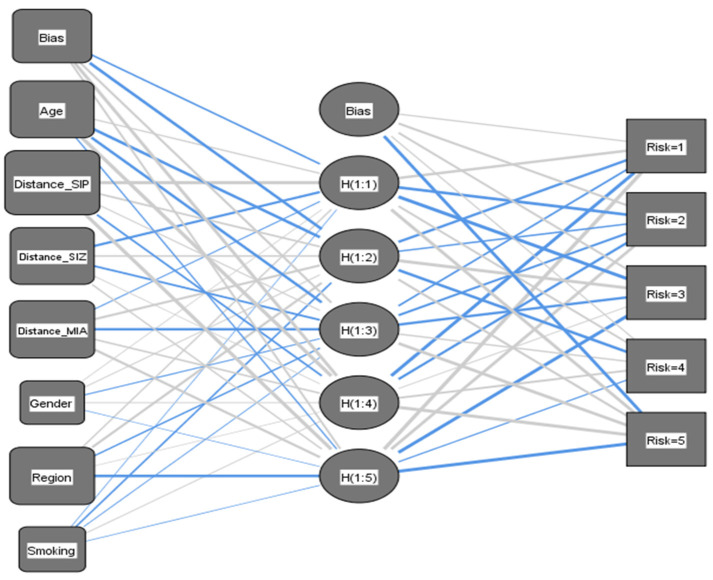
Multilayer perceptron—architecture of neural network.

**Figure 10 ijerph-21-00553-f010:**
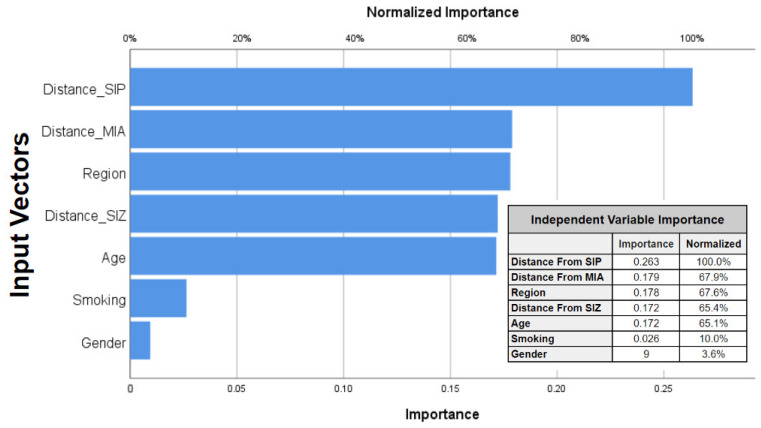
Normalized importance analysis of input vectors. The inset table provides a comprehensive overview of the importance and corresponding normalized importance values for different indicators.

**Table 1 ijerph-21-00553-t001:** Descriptive statistics of the relationship between exposure zones and asthma severity. Characteristics of asthma severity levels (severe, moderate, and mild) in the studied population (all) across exposure zones near SIP. Case numbers and their percentages among asthma severity levels and among different exposure zones are shown.

	Exposure Zone	Total	Severe	Moderate	Mild
		(Exposure Zones %)	Case Number (Asthma Severity %–Exposure Zones %)		
All	High (<6 km)	210 (51.2%)	56 (26.7%–51.9%)	107 (51%–49.1%)	47 (22.4%–56%)
	Middle (6–12 km)	146 (35.6%)	38 (26%–35.2%)	76 (52.1%–34.9%)	32 (21.9%–38.1%)
	Low (>12 km)	54 (13.2%)	14 (25.9%–13%)	35 (64.8%–16.1%)	5 (9.3%–6%)
Male	High (<6 km)	105 (55.6%)	31 (29.5%–63.3%)	52 (49.5%–49.5%)	22 (21%–62.9%)
	Middle (6–12 km)	61 (32.3%)	14 (23%–28.6%)	35 (57.4%–33.3%)	12 (19.7%–34.3%)
	Low (>12 km)	23 (12.2%)	4 (17.4%–8.2%)	18 (78.3%–17.1%)	1 (4.3%–2.9%)
Female	High (<6 km)	105 (47.5%)	25 (23.8%–42.4%)	55 (52.4%-48.7%)	25 (23.8%–51%)
	Middle (6–12 km)	85 (38.5%)	24 (28.2%–40.7%)	41 (48.2%–36.3%)	20 (23.5%–40.8%)
	Low (>12 km)	31 (14%)	10 (32.3%–16.9%)	17 (54.8%–15%)	4 (12.9%–8.2%)

**Table 2 ijerph-21-00553-t002:** The association between severe asthma cases and different industrial zones in the north of Sohar city. Asthma patients were categorized into three exposure zones based on their proximity to the pollution areas: SIP, SIZ, and MIA. A control area (CA) was selected as a reference group. The table displays estimated risk ratios (RRs) with confidence intervals (CI^95%^) and Chi-square test values (X^2^) for the high-exposure zone, comparing the development of severe asthma symptoms to patients with moderate and mild asthma in the low-exposure zone. The total number of cases and the number of cases compared between the different industrial zones are shown. The significance of the differences was determined by a Mann–Whitney U-test at *p* < 0.05 *.

	Valid Samples	High	Low	Severe Asthma (High/Low)	Moderate and Mild Asthma (High/Low)	RR (CI95%)	X^2^
SIP	264	<6 km	>12 km	47/5	163/49	2.42 (1.01–5.78)	0.031 *
SIZ	264	<10 km	>20 km	40/11	122/64	1.68 (0.92–3.09)	0.081
MIA	176	<6 km	>12 km	27/11	72/66	1.91 (1.01–3.6)	0.038 *
CA	323	<8 km	>12 km	25/41	118/139	0.77 (0.49–1.2)	0.241
SIP—North	223	Middle	North	23/20	82/98	1.29 (0.75–2.21)	0.349
SIP—South	269	Middle	South	23/40	82/124	0.9 (0.57–1.41)	0.639

**Table 3 ijerph-21-00553-t003:** Risk ratios of different regions for developing severe asthma symptoms. The table presents risk ratio and Chi-square test values (X^2^) for various regions in relation to the development of severe asthma symptoms compared to patients with combined moderate and mild asthma in control regions (>12 km) from SIP. The distances from SIP, SIZ, and MIA, as well as the number of cases (N) for each region, are shown. The significance of the differences was determined by a Mann–Whitney U-test at *p* < 0.05 *.

	Distance from SIP (km)	Distance from SIZ (km)	Distance from MIA (km)	Total *N*	Severe Asthma *N*	Moderate and Mild Asthma *N*	Severe Asthma %	Latitude	Longitude	RR (CI95%)	χ^2^
Ghadhfan	2.6	11.3	5.3	30	8	22	27%	24.471	56.602	2.88 (1.03–8.02)	0.035 *
Al Hadd	3.4	13.8	6.2	21	5	16	24%	24.487	56.585	2.57 (0.83–7.98)	0.10
Liwa	4.0	16.0	8.8	22	5	17	23%	24.512	56.586	2.45 (0.79–7.65)	0.12
Harmul	4.1	16.7	10.3	21	5	16	24%	24.523	56.596	2.57 (0.83–7.98)	0.10
Al Khuwayriyyah	4.7	8.0	6.1	12	2	10	17%	24.449	56.626	1.8 (0.4–8.2)	0.45
Al Eqdah	5.1	15.7	7.2	13	3	10	23%	24.498	56.569	2.49 (0.68–9.12)	0.17
Mikhaylif	5.1	16.5	8.4	19	2	17	11%	24.508	56.572	1.14 (0.24–5.38)	0.87
Majees	6.0	8.2	9.6	72	17	55	24%	24.455	56.657	2.55 (1–6.48)	0.036 *
Harat Ash Shaykh	6.1	6.8	7.6	8	2	6	25%	24.442	56.641	2.7 (0.63–11.65)	0.19
Falaj Al Qabail	6.4	6.4	5.9	69	19	50	28%	24.433	56.626	2.97 (1.19–7.45)	0.011 *
Az Zahiyah	7.7	19.7	11.5	40	5	35	13%	24.537	56.563	1.35 (0.42–4.35)	0.61
Nabr	10.1	22.5	14.5	20	5	15	25%	24.563	56.559	2.7 (0.87–8.35)	0.08
Rumelah	10.7	23.1	15.4	7	1	6	14%	24.572	56.563	1.54 (0.21–11.37)	0.67
Control	19.0	26.0	18.8	54	5	49	9%	24.505	56.499	-	-

**Table 4 ijerph-21-00553-t004:** **Descriptive statistics of the relationship between patients’ ages and asthma severity.** Distribution of asthma patients from the different types (severe, moderate, and mild) based on three age ranges (<25, 25–50 and >50) in male and female patients in total compared to high-exposure zone (<6 km from SIP). Estimated risk ratios (RRs) with confidence intervals (CI^95%^) and Chi-square test values (X^2^) of the higher age ranges (25–50 and >50) to develop severe asthma symptoms compared to lower age range (<25) with combined asthma (moderate and mild). The significance of the differences was determined by a Mann–Whitney U-test at *p* < 0.05 *.

Exposure Zone	Age Range	Total	Severe	Moderate	Mild	All		Male		Female	
		Age Ranges (%)	Case Number (Asthma Severity %–Age Range %)			RR (CI95%)	χ^2^	RR (CI95%)	χ^2^	RR (CI95%)	χ^2^
Total	<25	150 (36.6%)	20 (13.3%–23.8%)	81 (54%–37.2%)	49 (32.7%–45.4%)	-	-	-	-	-	-
	25–50	150 (36.6%)	41 (27.3%–48.8%)	82 (54.7%–37.6%)	27 (18%–25%)	2.05 (1.26–3.33)	0.003 *	2.8 (1.4–5.58)	0.002 *	1.417 (0.72–2.80)	0.304
	>50	110 (26.8%)	23 (20.9%–27.4%)	55 (50%–25.2%)	32 (29.1%–29.6%)	1.57 (0.91–2.71)	0.104	1.97 (0.86–4.5)	0.11	1.14 (0.54–2.4)	0.727
High <6 km from SIP	<25	83 (39.5%)	14 (16.9%–29.8%)	42 (50.6%–39.3%)	27 (32.5%–48.2%)	-	-	-	-	-	-
	25–50	80 (38.1%)	24 (30%–51.1%)	42 (52.5%–39.3%)	14 (17.5%–25%)	1.78 (0.99–3.19)	0.047 *	2.81 (1.23–6.4)	0.1	1.058 (0.47–2.38)	0.892
	>50	47 (22.4%)	9 (19.1%–19.1%)	23 (48.9%–21.5%)	15 (31.9%–26.8%)	1.14 (0.53–2.42)	0.743	1.286 (0.37–4.46)	0.694	0.857 (0.33–2.24)	0.753

**Table 5 ijerph-21-00553-t005:** The link between patients’ smoking state and asthma severity. Descriptive statistics of the smoking status (smoker, passive smoker, and non-smoker) in male and female patients in total compared to the high-exposure zone (<6 km from SIP) and the relationship with the asthma severity levels (severe, moderate, and mild, *) are shown. Estimated risk ratios (RRs) with confidence intervals (CI95%) and Chi-square test values (χ^2^) of the different smoking states (smokers and passive smokers) to develop severe asthma symptoms compared to non-smoker patients with mild or combined asthma (moderate and mild, **) are also presented.

Exposure Zone	Smoking State	All	Male/Female	Severe *	Moderate *	Mild *	Severe/Mild		Severe/Comb **	
		(Smoking Groups %)		Case Number (Asthma Severity %–Smoking Groups %)			RR (CI 95%)	χ^2^	RR (CI 95%)	χ^2^
Total	Non- Smoker	347 (84.6%)	155/192 (82.0%–86.9%)	71 (20.5%–84.5%)	178 (51.3%–81.6%)	98 (28.2%–90.7%)	-	-	-	-
	Smoker	22 (5.4%)	19/3 (10.1%–1.4%)	5 (22.7%–5.9%)	15 (68.2%–6.9%)	2 (9.1%–1.8%)	1.7 (1.03–2.81)	0.12	1.11 (0.5–2.47)	0.80
	Passive Smoker	41 (10%)	15/26 (7.9%–11.8%)	8 (19.5%–9.5%)	25 (61%–11.5%)	8 (19.5%–7.4%)	1.19 (0.71–2)	0.54	0.95 (0.5–1.84)	0.89
High <6 km from SIP	Non- Smoker	179 (85.2%)	89/90 (84.8%–85.7%)	43 (24%–91.5%)	85 (47.5%–79.4%)	51 (28.5%–91.1%)	-	-	-	-
	Smoker	9 (4.3%)	8/1 (7.6%–1.0%)	1 (11.1%–2.1%)	7 (77.8%–6.5%)	1 (11.1%–1.8%)	1.09 (0.27–4.45)	0.90	0.46 (0.07–2.99)	0.37
	Passive Smoker	22 (10.5%)	8/14 (7.6%–13.3%)	3 (13.6%–6.4%)	15 (68.2%–14%)	4 (18.2%–7.1%)	0.94 (0.39–2.27)	0.88	0.57 (0.19–1.68)	0.27

**Table 6 ijerph-21-00553-t006:** Relationship between exposure to contaminants and asthma severity. Descriptive statistics regarding the levels of exposure to various contaminants (dust, incense, perfume, and smoke) and their corresponding asthma severity levels. The table provides estimated risk ratios (RRs) with confidence intervals (CI^95%^) and Chi-square test values (X^2^) compared to non-exposed patients with combined asthma (moderate and mild). The analysis is conducted for both the entire study area and the high-exposure zone within a radius of less than 6 km from the port.

Contaminants	Respondent *N*	Respondent %	Responses %	Severe Asthma %	Severe Asthma Y/N	Comb Asthma Y/N	Total		High (<6 km)	
							RR (CI95%)	χ^2^	RR (CI95%)	χ^2^
Yes	281	68.5%	-	19.2%	54	227	0.83 (0.56–1.23)	0.347	0.76 (0.45–1.28)	0.303
Uncertain	129	31.5%	-	23.3%	30	99	-	-	-	-
Dust	186	67.9%	27.7%	22.6%	42/42	144/182	1.2 (0.82–1.76)	0.339	1.02 (0.61–1.69)	0.951
Incense	224	81.8%	33.4%	20.1%	45/39	179/147	0.96 (0.65–1.4)	0.826	1 (0.6–1.67)	0.99
Perfume	180	65.7%	26.8%	21.7%	39/45	141/185	1.11 (0.76–1.62)	0.601	1.21 (0.73–2.01)	0.457
Smoke	81	29.6%	12.1%	24.7%	20/64	61/265	1.27 (0.82–1.97)	0.295	1.41 (0.81–2.47)	0.238
Total Respondents	274	100.0%	-	19.3%	53	221	-	-	-	-
Total Responses	671	-	100.0%	21.8%	146	525	-	-	-	-

**Table 7 ijerph-21-00553-t007:** Classification of percentage of prediction. The actual and expected number of samples based on the model, as well as the percentages of the correct predictions for each level of risk. The data are reported separately for the training and testing phases of constructing the neural network.

		Predicted					
		Very Low Risk	Low Risk	Average Risk	High Risk	Very High Risk	Percent Correct
Training	Very Low Risk	47	1	0	0	0	97.9%
	Low Risk	1	59	4	0	0	92.2%
	Average Risk	0	1	53	1	3	91.4%
	High Risk	0	0	0	57	2	96.6%
	Very High Risk	0	0	1	1	55	96.5%
	Overall Percent	16.8%	21.3%	20.3%	20.6%	21.0%	94.8%
Testing	Very Low Risk	29	0	0	0	0	100.0%
	Low Risk	1	24	1	1	1	88.9%
	Average Risk	0	0	17	2	3	77.3%
	High Risk	0	0	1	18	2	85.7%
	Very High Risk	17.7	0	0	1	24	96.0%
	Overall Percent	23.4%	19.4%	15.3%	17.7%	24.2%	90.3%

## Data Availability

The manuscript comprehensively incorporates the majority of the data generated or scrutinized throughout this study. Should further specific data be required, it can be promptly supplied upon request, subject to mutual agreement.

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
