# Peer review of "The Impact of Air Pollution on Asthma Severity among Residents Living near the Main Industrial Complex in Oman: A Cross-Sectional Study"

_ijerph, 2024, doi:10.3390/ijerph21050553_

Round 1
Reviewer 1 Report
Comments and Suggestions for Authors
The introduction needs to be shortened, i.e. the part of the text that is not related to the connection between asthma and air pollution should be removed.
In the method, it is necessary to specify the inclusion and exclusion criteria, for example, for the diagnosis of asthma and for the exclusion of comorbid diseases.
It is also necessary to state during which period the patients were classified according to the degree of severity of asthma.
It is necessary to clarify why the proportion of patients with severe asthma of 20% was taken in the calculation of the sample size (usually it is taken up to 10%).
In the results, Figure 1 needs to be refined, and percentages should be displayed, not just the absolute numbers of the columns. The height of the column with the same absolute number, e.g. 49 subjects with severe asthma, is lower than the height of the column with the same number of subjects (49) with mild asthma.
In the discussion, should comment the difference between women and men in the relationship between the severity of asthma and the average of distances of patients' residences from SIP (Figure 3).
It is necessary to emphasize the scientific contribution of this research, as well as its shortcomings.
Comments on the Quality of English Language
The quality of the English language is satisfactory, minor corrections are needed.
Author Response
Responses to Reviewer 1's Comments and Suggestions
- The introduction needs to be shortened, i.e. the part of the text that is not related to the connection between asthma and air pollution should be removed.
Our response: Thank you for your feedback, which has been invaluable in sharpening our manuscript's focus. We have trimmed the asthma section to emphasize its connection to air pollution while retaining essential information on prevalence, impact, and management in Oman
- In the method, it is necessary to specify the inclusion and exclusion criteria, for example, for the diagnosis of asthma and for the exclusion of comorbid diseases.
Our response: The inclusion and exclusion criteria are delineated within the ”Study Population” section of the methodology, written in red for emphasis. This addition has been highlighted in yellow for clarity.
- It is also necessary to state during which period the patients were classified according to the degree of severity of asthma.
Our response: The period during which patients were classified according to the degree of asthma severity is detailed in the “Asthma Severity Assessment” section of the methodology, highlighted in yellow for clarity. Here, we specify that this assessment was conducted prospectively during patients' health center visits, under the supervision of qualified physicians. Classification relied on a comprehensive evaluation, incorporating both medical record review and symptom assessment.
- It is necessary to clarify why the proportion of patients with severe asthma of 20% was taken in the calculation of the sample size (usually it is taken up to 10%).
Our response: Thank you for your valuable comment. We considered the incidence of severe asthma among Omani asthmatic patients, not the entire Omani population, estimating it at approximately 20% based on prior studies. This led to a sample size of 246, which is less than the number utilized in our study. However, using a 10% proportion, indicative of severe asthma prevalence in the broader Omani society, would result in a smaller estimated sample size of about 139 patients. This approach strengthens the validity and reliability of our study findings, allowing for more precise conclusions regarding the influence of asthma severity on our outcomes.
- In the results, Figure 1 needs to be refined, and percentages should be displayed, not just the absolute numbers of the columns. The height of the column with the same absolute number, e.g. 49 subjects with severe asthma, is lower than the height of the column with the same number of subjects (49) with mild asthma.
Our response: Thank you for your feedback regarding Figure 1. We have indicated in the description of the figure that the numbers above the bars represent the total number of patients in each category, while the height of the columns reflects the percentage of each level of asthma (High, Moderate and Severe) within each group of asthmatic patients. These percentages can be estimated through the Y-axis to the left. The primary aim of the figure is to demonstrate the uniform distribution of asthma severity levels between males and females, mirroring the general distribution among all asthma patients, with approximately 25% classified as mild, 55% as moderate, and 20% as severe.
- In the discussion, should comment on the difference between women and men in the relationship between the severity of asthma and the average of distances of patients' residences from SIP (Figure 3).·
Our response: Thank you for your input regarding the discussion on the relationship between asthma severity, gender, and proximity to the SIP. We have incorporated the comment about the disparity between women and men in this relationship in the discussion section, as suggested.
- It is necessary to emphasize the scientific contribution of this research, as well as its shortcomings.
Our response: Thank you for your valuable feedback. Our study makes significant scientific contributions by providing novel insights into the relationship between industrial pollution exposure and asthma severity in Oman. By identifying localized differences in pollution exposure and demographic vulnerabilities, we contribute to a deeper understanding of the complex interactions between environmental factors and respiratory health. However, we acknowledge limitations such as the use of cross-sectional data and the focus on a specific demographic and geographic area, which may impact the generalizability of our findings. Despite these shortcomings, our research provides valuable insights that can inform future studies and interventions aimed at mitigating the impact of pollution on respiratory health. This clarification has been added at the end of the discussion section highlighted in yellow color.
Comments on the Quality of English Language
- The quality of the English language is satisfactory, minor corrections are needed.
Our response: The language was reviewed, as you mentioned, and some previous linguistic lapses were corrected.
Reviewer 2 Report
Comments and Suggestions for Authors
Please see attachdes document for comments.

Suggest that writing could be much more succinctly and factual. This is preferred writing style with scientific manuscripts. An example of such a sentence is as follows:
"This classification framework provided a comprehensive understanding of the range of asthma severity experienced by the studied participants, enabling a meaningful analysis of the impact of air pollution on their asthma symptoms."
Author Response
Responses to Reviewer 2's Comments and Suggestions
- The authors address a major public health concern with significant immediate and longterm consequences.
Major recommendations / concerns
- Sample representation:
- The manuscript would benefit from a clear explanation of how the sample of 410 asthmatic patients was obtained. Understanding the sample selection process is crucial for assessing the study's validity. It is imperative to provide detailed information on whether this was a convenience sample or if specific criteria were used for.
Our response: Thank you for your valuable comment. Our study's sample selection process was meticulously designed to ensure robust representation of the asthmatic population in our targeted region and timeframe. We initially estimated the incidence of severe asthma among Omani asthmatic patients to be approximately 20%, leading to a calculated sample size of 246 individuals. However, in our study, we surpassed this figure by collecting data from all asthma patients who visited specified health centers in Al Batinah North Governorate between 2014 and 2022. This comprehensive approach allowed us to include a total of 410 patients, exceeding our initial calculation. Importantly, our sampling methodology was not based on convenience; instead, we employed specific criteria to capture data from all asthma patients within our designated region and timeframe. By providing these details, we aim to enhance transparency and facilitate a more thorough assessment of the study's validity. Clarification about this estimation is included in the “Sample Size Determination” section of the methodology.
- Selection of surrounding areas:
- The method used to select the 17 different surrounding areas of ASAP is not well-explained, while there is a reference to specific criteria ensuring representation and homogeneity, further clarification is needed. It is essential to explain how these areas were chosen to ensure transparency and reproducibility.
Our response: The selection of the 17 surrounding areas was guided by the placement of healthcare centers near the Sohar Industrial Port (SIP), catering to asthma patients from nearby regions. These centers were strategically chosen due to their proximity to the SIP and their function as primary healthcare providers for residents in adjacent areas. By incorporating data from these centers, our aim was to create a comprehensive dataset representing asthma patients affected by industrial pollution near the SIP. Patient numbers varied across these centers, reflecting differences in population density. Furthermore, we consolidated data from additional regions located beyond a 15-kilometer radius into a singular control group for comparative purposes. This methodology ensured both geographic diversity and adequate representation in our sample, encompassing the broader population affected by industrial activities. This explanation has been incorporated into the ”Study Population” section of the methodology section to provide clarity and transparency regarding our sampling approach.
- Asthma severity assessment:
- The manuscript should clarify if the physician's assessment was conducted retrospectively by reviewing case records. Additionally, the method used to confirm asthma diagnosis should be clearly stated, specifying whether it was solely based on a doctor’s diagnosis or if other diagnostic criteria were used.
Our response: Thank you for your insightful comments on our study's asthma severity assessment. We've updated the "Asthma Severity Assessment" in the methodology to clarify that the assessment was conducted during patients' visits to health centers by qualified physicians, adhering to standardized criteria by the Global Initiative for Asthma (GINA). This approach involved evaluating symptoms, medical history, and severity classification into three levels: severe, moderate, and mild. This ensures a comprehensive understanding of asthma severity and enhances the analysis of air pollution's impact on participants' symptoms.
- Exposure zones:
- The rationale behind determining the specific exposed zones, such as the 6-kilometer radius, is unclear. Understanding how this conclusion was reached is essential as this could impact the classification categories. This was deliberated on in the article published by
Our response: We appreciate your inquiry about how we determined the specific exposure zones in our study. Asthma patients were categorized into three exposure zones (high, medium, and low) based on their proximity to the Sohar industrial port. To achieve this, we built upon a previously published study by Al-Wahaibi and Zeka (2015), which investigated the health impacts from living near a major industrial park in Oman, making slight adjustments to achieve a more balanced distribution of patient samples among the regions. This approach allowed us to better capture the varying levels of pollution exposure experienced by residents in proximity to the industrial complex.
Al-Wahaibi, A., & Zeka, A. (2015). Health impacts from living near a major industrial park in Oman. BMC public health, 15, 1-10.
- Time period of study:
- It is unclear during which time period the study was conducted Previous reports suggest that pollution may have seasonal influences, indicating the importance of specifying the study duration and considering seasonal variations in pollutant levels. The potential impact of environmental factors, like wind direction,on pollutant dispersion should be considered as discussed in the article authored by Al Wahaibi 2015.
Our response: Your observation is very important, but unfortunately, we lack the detailed temporal data for the samples that would enable us to analyze this concept thoroughly.
Minor recommendations
- Handling of missing data:
- Additional information is needed regarding how participants with missing data were handled in the analysis. Understanding the approach to dealing with missing data is essential to assess the potential for selection bias.
Our response: Your suggestion has been implemented, and the information regarding the handling of missing data has been added to the "Sample Size Determination" section in the methodology, highlighted in yellow color. Your input has been invaluable in improving the transparency and thoroughness of our study methodology.
- Consideration of confounding factors:
- The manuscript should consider other confounding factors, such as socioeconomic status, that could impact socioeconomic factors, would enhance the comprehensiveness of the analysis. asthma diagnosis or exacerbation. Providing a clear description of the area under study, including socioeconomic factors, would enhance the comprehensiveness of the analysis.
Our response: Thank you for your valuable feedback. We recognize the importance of considering socioeconomic status in our analysis for a more thorough understanding of industrial pollution's impact on asthma outcomes. While we initially intended to include factors like income, education, occupation, housing and living conditions, and healthcare access, incomplete patient records hindered our ability to analyze and link these variables. With missing data surpassing 30%, deriving statistically significant insights became challenging. We have addressed these limitations in our discussion, emphasizing the impact of missing data on the comprehensiveness of our analysis (highlighted in yellow color).
Comments on the Quality of English Language
- Suggest that writing could be much more succinctly and factual. This is preferred writing style with scientific manuscripts. An example of such a sentence is as follows:
"This classification framework provided a comprehensive understanding of the range of asthma severity experienced by the studied participants, enabling a meaningful analysis of the impact of air pollution on their asthma symptoms."
Our response: Thank you for highlighting the need for a more concise and factual writing style in our manuscript. We have already addressed this concern and have revised the text accordingly. We appreciate your feedback and strive to ensure clarity and precision in our scientific communication.
Reviewer 3 Report
Comments and Suggestions for Authors
This paper is interesting in that it provides extensive information on asthma affecting population living near sources of atmospheric pollution. It is a pity that the nymber of subjects investigated is not very high; however, it was sufficient to reinforce the conclusions. Anyway, methods used can be easily adopted by other authors to evaluate health effects by pollution in other areas. For that, the paper is suitable publication, providing that the following adds-on and changes are made.
1) it would be important to add in the introduction some data about the pollution levels measured and/or expected in the study regions. Synthetic data are sufficient.
2) Legends for tables and figures should be concise. Comments about the content should be reported in the text and not in the legends.
Comments on the Quality of English LanguageThere are minor flaws in language (as an example, the first line of legend of table 3) or two lines before fig7 where wield should be replaced by yield. Please check again the text.
Author Response
Responses to Reviewer 3's Comments and Suggestions
- This paper is interesting in that it provides extensive information on asthma affecting population living near sources of atmospheric pollution. It is a pity that the nymber of subjects investigated is not very high; however, it was sufficient to reinforce the conclusions. Anyway, methods used can be easily adopted by other authors to evaluate health effects by pollution in other areas. For that, the paper is suitable publication, providing that the following adds-on and changes are made.
Our response: We are pleased to express our deep appreciation for your valuable scientific opinion.
- 1) it would be important to add in the introduction some data about the pollution levels measured and/or expected in the study regions. Synthetic data are sufficient.
Our response: Thank you for your valuable input. We have incorporated the air quality data you mentioned into the introduction of our manuscript, providing a more comprehensive overview of the environmental context surrounding our study. This addition has been highlighted in yellow for clarity.
- 2) Legends for tables and figures should be concise. Comments about the content should be reported in the text and not in the legends.
Our response: Your attention to detail is greatly appreciated. We have made the necessary adjustments to the legends for tables and figures, streamlining them to include only essential information. Additionally, the relevant comments about the content have been appropriately included in the main text in the methodology. These modifications have been highlighted in yellow for clarity.
Comments on the Quality of English Language
- There are minor flaws in language (as an example, the first line of legend of table 3) or two lines before fig7 where wield should be replaced by yield. Please check again the text.
Our response: The language was reviewed, as you mentioned, and some previous linguistic lapses were corrected.